# GaussianTalker: Real-Time High-Fidelity Talking Head Synthesis with Audio-Driven 3D Gaussian Splatting

## ABSTRACT

This paper proposes GaussianTalker, a novel framework for real-time generation of pose-controllable talking heads. It leverages the fast rendering capabilities of 3D Gaussian Splatting (3DGS) while addressing the challenges of directly controlling 3DGS with speech audio. GaussianTalker constructs a single 3DGS representation of the head and deforms it in sync with the audio. A key insight is to encode the 3D Gaussian attributes into a shared implicit feature representation, where it is merged with audio features to manipulate each Gaussian attribute. This design exploits the spatial information of the head and enforces interactions between neighboring points. The feature embeddings are then fed to a spatial-audio attention module, which predicts frame-wise offsets for the attributes of each Gaussian. This method is more stable than previous concatenation or multiplication approaches for manipulating the numerous Gaussians and their intricate parameters. Overall, GaussianTalker offers a promising approach for real-time generation of high-quality pose-controllable talking heads.

## CCS CONCEPTS

• **Computing methodologies** → **Reconstruction**; 3D imaging; • **Information systems** → *Multimedia content creation*.

## KEYWORDS

Talking Head Generation, 3D Controllable Head, 3D Gaussian Splatting

## 1 INTRODUCTION

Generating a talking head video driven by arbitrary speech audio is a popular task that has various uses, including the generation of digital humans, virtual avatars, movie production, and teleconferencing [6, 20, 32, 35, 37, 39, 42, 53]. While various works [6, 20, 32, 42] have successfully attempted to solve this task using generative models, they do not focus on controlling head poses, limiting their realism and applicability. Recently, numerous studies [16, 23, 26, 38, 47, 48] have applied neural radiance fields (NeRF) [30] for the creation of pose controllable talking portraits. By directly conditioning audio features in the multi-layer perceptron (MLP) of NeRF, these methods can synthesize view-consistent 3D head structure with its lips synced to the input audio. Although these NeRF-based techniques achieve high-quality and consistent visual outputs, their

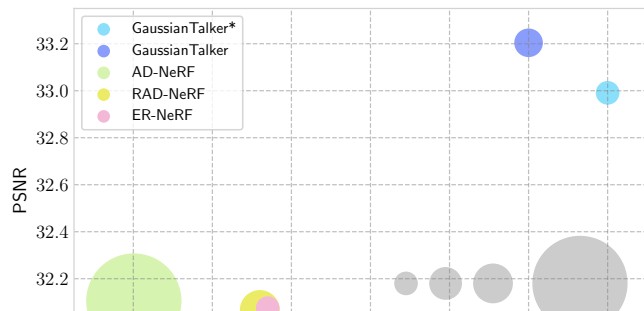

**Figure 1: Fidelity and inference time comparison between existing 3D talking face synthesis models [16, 23, 38] and ours. Our method, GaussianTalker, achieves on par with or better results at much higher FPS. Note that we also include GaussianTalker***, **a more efficient and faster variant. Size of each bubble represents the training time of each method.**

slow inference speed limits their practicality. Despite recent advancements [23, 38] achieving rendering speeds up to 30 frames per second (fps) at $512 \times 512$ resolution, computational bottlenecks must be overcome to be applied in real-world scenarios.

Addressing this limitation, an intuitive solution is to leverage the fast rendering capabilities of 3D Gaussian Splatting (3DGS) [21]. Recently recognized as a viable alternative to NeRF, 3DGS offers comparable rendering quality while significantly improving inference speeds. Although 3DGS was initially proposed for reconstructing static 3D scenes, subsequent works have extended it to dynamic scenes [29, 43–45]. However, there has been little research on leveraging 3DGS to create dynamic 3D scenes with controllable inputs, most of which focused on using an intermediate mesh representation to drive the 3D Gaussians [7, 18, 25, 27, 33]. However, relying on an intermediate 3D mesh representation, such as FLAME [24], for deformation often lacks fine details in hair and facial wrinkles.

We identify two major challenges in directly mapping the speech audio to the deformation of 3D Gaussians. First, the 3DGS representation lacks shared spatial information among the adjacent points, complicating its manipulation. The optimization process of 3DGS does not consider relationships between neighboring Gaussians, crucial for maintaining facial region cohesion during deformation. Secondly, the extensive parameter space and a substantial number of Gaussians pose a challenge to their manipulation. Unlike controllable NeRF representations where the position and the number of sampling points are fixed, the position, shape, and appearance attributes of numerous Gaussian points need to be deformed per frame, while also preserving the intricate facial details.

In this paper, we present **GaussianTalker**, a novel framework for real-time pose-controllable talking head synthesis. For the first time, we leverage the 3D Gaussian representation to exploit its fast scene modeling capability for audio-driven dynamic facial animation. We construct a static 3DGS representation of the canonical head shape and deform this in sync with the audio. Specifically, we employ a multi-resolution triplane to extract feature embeddings for each 3D Gaussian position, from which each Gaussian attribute is directly estimated. This design ensures that the triplane learns the spatial and semantic information of the 3D head, while the interpolation mechanism of the 2D feature grids efficiently enforces interactions between neighboring points. The feature embeddings are subsequently fed to the proposed spatial-audio attention module, where they are merged with the audio features to predict the frame-wise offsets for the attributes of each Gaussian. This module successfully models the relevance between audio features and the motions for each Gaussian primitive. The cross attention offers a more stable approach of manipulating the substantial number of Gaussians and their intricate parameter space, compared to concatenation [16, 38] or multiplication [23] as in previous works. Qualitative and quantitative experiments demonstrate GaussianTalker's superiority in facial fidelity, lip synchronization accuracy, and rendering speed compared to previous methods. Additionally, we conduct ablation studies to verify the effectiveness of individual design choices within our model.

Our main contributions are summarized as follows:

- For the first time, we present a novel audio-conditioned 3D Gaussian Splatting framework real-time 3D-aware talking head synthesis.
- We reformulate the 3D Gaussian representation with a feature volume representation in order to enforce spatial consistency among adjacent Gaussians.
- We integrate cross-attention mechanisms between audio and spatial features to improve stability and ensure region-specific deformation across a significant number of Gaussians.

## 2 RELATED WORK

### 2.1 Audio-driven talking portrait synthesis

Audio-driven talking portrait synthesis aims to create realistic facial animations with accurate lip movements based on audio input. Early 2D GAN-based methods [32, 36, 50, 51, 57] achieved photorealism but lacked control over head pose due to the absence of 3D geometry. In order to control the head poses, some works [28, 39, 41, 53] utilize model-based methods, where facial landmarks and 3D morphable models reinforce the lip sync model with the ability to adjust the orientation of the head. However, these approaches lead to new problems such as extra errors from the intermediate representations, and inaccuracies in identity preservation and realism.

Recently, Neural Radiance Fields (NeRF) [30] have been explored for talking portraits due to their ability to capture complex scenes. AD-NeRF [16] pioneered using NeRF's implicit representation for conditional audio input, but separate networks for head and torso limited its flexibility. Subsequent NeRF-based methods [26, 34, 46] achieved high quality but suffered from slow rendering speeds. While RAD-NeRF [38] and ER-NeRF [23] improved efficiency and

quality with grid-based NeRF [31], real-time rendering of pose-controllable 3D talking head remains challenging.

### 2.2 3D Gaussian splatting

3DGS [21] is a pioneering technique in point cloud rendering that utilizes a multitude of ellipsoidal, anisotropic balls to precisely represent a scene. Each point embodies a 3D Gaussian distribution, with its mean, covariance, opacity, and spherical harmonics parameters optimized to accurately capture the scene's shapes and appearances. This approach effectively resolves common issues in point rendering, such as output gaps. Furthermore, combined with a tile-based rasterization algorithm, it facilitates expedited training and real-time rendering capabilities. Recently, 3DGS has gained widespread application in 3D vision tasks such as object manipulation [10, 13], reconstruction [11, 21], and perception [4, 29] within 3D environments.

### 2.3 Facial animation with 3DGS

Previous methods for facial reconstruction and animation primarily relied on 3D Morphable Models(3DMM) [15, 22] or utilized neural implicit representations [1, 14, 55]. Recent approaches [7, 8, 33, 40] have shifted towards adopting the 3DGS representation, aiming to leverage the benefits of rapid training and rendering while still achieving competitive levels of photorealism. GaussianAvatars [33] reconstructed head avatars by rigging 3D Gaussians on FLAME [24] mesh. MonoGaussianAvatar [7] learned explicit head avatars by shifting the mean position of 3D Gaussians from canonical to deformed space using Linear Blend Skinning (LBS) and simultaneously adjusts other Gaussian parameters through a deformation field. GaussianHead [40] adopted a motion deformation field to adapt to facial movements while preserving head geometry and separately utilized a tri-plane to retain the appearance information of individual 3D Gaussians. However, the aforementioned methods tend to depend on parametric models for facial animation. In contrast to previous works, our audio-driven method is not only free from the need for data beyond the speech sequence for facial reenactment but also is readily applicable to novel audio.

## 3 PRELIMINARY: 3D GAUSSIAN SPLATTING

3D Gaussian splatting (3DGS) [21] employs anisotropioc 3D Gaussians as geometric primitives for learning an explicit 3D representation. Each 3D Gaussian is defined by a center mean $\mu \in \mathbb{R}^3$ and covariance matrix $\Sigma \in \mathbb{R}^{3\times3}$ in the 3D coordinate as follows:

$$g(x) = \exp\left(-\frac{1}{2}(x - \mu)^T \Sigma^{-1}(x - \mu)\right), \quad (1)$$

for a 3D coordinate $x \in \mathbb{R}^3$. The covariance matrix $\Sigma$ is further decomposed into $\Sigma = RSS^T R^T$ with a scaling matrix $S$ and a rotation matrix $R$, defined by a scaling factor $s \in \mathbb{R}^3$ and a learnable quaternion $r \in \mathbb{R}^4$, respectively. Additionally, to encode the appearance information, each 3D Gaussian contains a set of spherical harmonics with degree $k$ such that $SH \in \mathbb{R}^{3(k+1)(k+1)}$, along with an opacity value $\alpha \in \mathbb{R}$. In summary, 3DGS represents a 3D scene with a set of 3D Gaussians parameters, defined as:

$$\mathcal{G} = \{\mu, r, s, SH, \alpha\}, \quad (2)$$

**Figure 2: Overview of our GaussianTalker framework. GaussianTalker utilizes a multi-resolution triplane to leverage different scales of features depicting a canonical 3D head. These features are fed into a spatial-audio attention module along with the audio feature to predict per-frame deformations, enabling fast and reliable talking head synthesis.**

Given a novel viewing direction $\pi$, a 2D image $\hat{I}$ is rendered as:

$$\hat{I} = \mathcal{R}(\mathcal{G}; \pi), \tag{3}$$

where $\mathcal{R}(\cdot)$ is the differentiable rasterizer.

More specifically, for $\mathcal{R}(\cdot)$, 3DGS employs differential splatting [49] during novel view rendering. In order to project 3D Gaussians to 2D for rendering, the covariance matrix in the 2D space, $\Sigma' \in \mathbb{R}^{2 \times 2}$, is calculated by viewing transform $W$ and the Jacobian $J$ of the affine approximation of the projective transformation [58], such as:

$$\Sigma' = JW\Sigma W^T J^T. \tag{4}$$

Subsequently, the color of each pixel is computed by blending all Gaussians that overlap the pixel and ordered by their depths as follows:

$$C = \sum_{i=1}^{i} c_i \alpha_i' \prod_{j=1}^{i-1}(1 - \alpha_j'), \tag{5}$$

where $c_i$ is the color of each point determined using the SH coefficient with view direction, and $\alpha_i'$ is computed by the multiplication of the opacity $\alpha$ of the 3D Gaussian and its projected covariance $\Sigma'$.

## 4 METHODOLOGY

### 4.1 Problem formulation and Overview

In this section, we describe the main components of **GaussianTalker**, designed for the real-time synthesis of high-fidelity, pose-controllable talking head images driven by audio input. Our model is trained on a talking portrait video $\mathcal{V} = \{I_n\}$ consisting of $N$ number of image frames for an identity. Our objective is to reconstruct a set of canonical 3D Gaussians that represent the mean shape of the talking head, and learn a deformation module that deforms the 3D Gaussians according to corresponding input audio. During inference, for the input audio $a_n$, the deformation module predicts the

offsets of each Gaussian attribute, and the deformed Gaussians are rasterized at the viewing point $\pi_n$ to output the novel image $\hat{I}_n$.

An overview of our proposed method is depicted in Fig. 2. We first introduce the multi-resolution tri-plane that encodes the low-dimensional features of the 3D Gaussians to represent the static mean shape of the canonical head in Sec. 4.2. In Sec. 4.3, we introduce the speech-motion cross-attention module that fuses 3D Gaussians features and audio features to accurately model facial motion driven by input audio. Finally, Sec. 4.4 describes the stage-wise training strategy and the utilized loss functions.

### 4.2 Learning canonical 3D Gaussians with triplane representation

In this section, we introduce the details of learning the canonical shape of the talking head with 3D Gaussian representation. The vanilla implementation of 3DGS [21] does not inherently capture the spatial relationships between neighboring and distant 3D Gaussians. However, an ideal feature representation for a dynamic 3D head should be analogous for proximal facial regions and distinct for separated ones, as the close facial primitives would likely move to the same direction.

To realize this, we modify the 3D Gaussian representation by learning a low-dimensional feature representation, which can be later merged with the audio features for per-Gaussian deformation. We formulate the embedding space to encode information of the attributes of the 3D Gaussians, in order to take into account the shape and appearance of each Gaussian when predicting its deformation offsets. More specifically, we adopt a hybrid 3D representation that utilizes the explicit 3D representation of 3DGS, while also taking advantage of the encoded spatial information of implicit neural radiance fields [30]. For each of the canonical 3D positions $\mu_c$, we extract feature embeddings $f(\mu_c)$ from a multi-resolution triplane

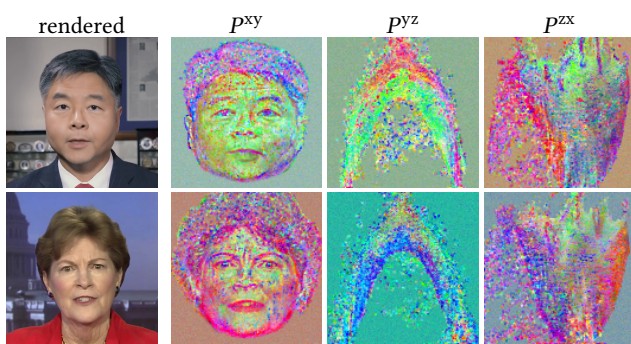

rendered $P^{xy}$ $P^{yz}$ $P^{zx}$

**Figure 3: Visualization of the triplane feature grids. The sequence displays a rendered image, followed by its orthographic projections: frontal (xy), overhead (yz), and side (zx) views.**

representation [3, 5, 12]. These feature embeddings are utilized to calculate the scale $s_c$, rotation $r_c$, spherical harmonics $SH_c$, and opacity $\alpha_c$ of each point. These computed attributes make up the canonical 3D Gaussian of the talking head, denoted as:

$$\mathcal{G}_{\text{can}} = \{\mu_c, r_c, s_c, SH_c, \alpha_c\}. \tag{6}$$

During training, instead of directly updating the 3D Gaussian attributes, the feature grids of the triplane and the attribute prediction networks are optimized. This allows for the feature embedding $f(\mu_c)$ to store the region-specific facial information of the canonical 3D head, while also enforcing spatial relationships between neighboring Gaussians. In the following, we introduce the formulation of each module in detail.

*4.2.1 Triplane representation for 3D Gaussian.* In order to encode the spatial information of the canonical 3D head, we adopt a multi-resolution triplane representation, constructed by three orthogonal 2D feature grids, $P = \{P^{xy}, P^{yz}, P^{zx}\}$. Each of these planes has shape $H \times R \times R$, where $H$ stands for the hidden dimension of features, and $R$ denotes the resolution of each dimension. For individual 3D Gaussian with position $\mu$, each of its coordinate values is normalized between $[0, R)$, and its corresponding features are computed by interpolating the point into a regularly spaced 2D grid for each plane. These features are combined using the Hadamard product $\prod$ for each plane, followed by concatenation $\bigcup$ along the different dimensions, to produce a final feature vector $f(\mu)$ of length $H$ for each of the canonical Gaussian position $\mu_c$, such as:

$$f(\mu) = \bigcup \prod_{p \in P} \text{interp}(p, \zeta_p(\mu_c)), \tag{7}$$

where $\zeta_p(\mu)$ denotes a projection of $\mu$ onto the $p$'th plane and 'interp' denotes bilinear interpolation of a point into the regularly spaced 2D grid. The visualization of features in our multi-resolution triplane is depicted in Fig. 3.

*4.2.2 Attribute prediction of canonical 3D Gaussians.* Unlike the original 3DGS implementation shown in (2), we do not explicitly store the shape information $r$ and $s$, and the appearance information $SH$ and $\alpha$. Instead, these attributes are obtained from the corresponding feature representation $f(\mu)$. Specifically, we employ a set of MLP layers, denoted as $\mathcal{F}_{\text{can}}(\cdot)$, to map the feature to the

mean scale $s_c$, mean rotation $r_c$, mean spherical harmonics $SH_c$, and mean opacity value $\alpha_c$ from $f(\mu)$, such as:

$$\{s_c, r_c, SH_c, \alpha_c\} = \mathcal{F}_{\text{can}}(f(\mu)). \tag{8}$$

Compared to the original 3DGS [21] where each Gaussian is optimized independently, our hybrid representation conditioned on an implicit feature volume enforces shared facial information between adjacent points.

## 4.3 Learning audio-driven deformation of 3D Gaussians

Previous works [16, 23, 26, 38] employ a conditional NeRF representation, wherein the 3D coordinates of the sampling point along each ray remain fixed, with only color and density conditioned to input audio. However, in order to fully benefit from the explicit representation of 3DGS, we choose to deform the 3D Gaussians, where we manipulate not only the appearance information but also the spatial positions and shape of each Gaussian primitive. While this can more accurately capture the constantly fluctuating 3D shape of the talking head, deformation of 3D Gaussians is a much more complex task compared to controlling a NeRF representation. The intricate nature of Gaussian primitives, coupled with their sheer quantity, presents significant challenges for deformation due to the extensive parameter space of 3D Gaussians. In addition, input audio does not impact the whole facial image uniformly, making it vital for the deformation module to understand how varying facial regions respond to audio conditions for authentic facial animation.

In order to model the relations between the dynamic features and the vast amount of 3D Gaussians, we fuse the input speech audio $a_n$ with the encoded feature $f(\mu_c)$ in an attention mechanism, in order to produce the audio-aware feature $h_n$ for the $n$-th image frame. The deformation offsets of each Gaussian attribute for subsequent frames are directly conditioned on the feature $h_n$. Finally, the deformed set of 3D Gaussian for the $n$-th image frame is defined as:

$$\mathcal{G}_{\text{deform},n} = \{\mu_c + \Delta\mu_n, r_c + \Delta r_n, s_c + \Delta s_n, SH_c + \Delta SH_n, \alpha_c + \Delta\alpha_n\}, \tag{9}$$

where $\Delta\mu_n, \Delta s_n, \Delta r_n, \Delta SH_n, \Delta\alpha_n$ are the deformation offsets at $n$-th frame for 3D position, scale, rotation, spherical harmonics parameters and opacity, respectively. The details of each module is introduced in the following paragraphs.

*4.3.1 Spatial-audio cross-attention.* Previous approaches to implement region-aware audio, like ER-NeRF [23], simply adjust the weights for the audio features at each 3D point through elementwise multiplication. However, it encounters a challenge in that, regardless of the diverse audio inputs in a dynamic scene, a particular static 3D point consistently maintains the same audio weight. This fails to acknowledge that a fixed 3D coordinate may not consistently correspond to the same facial region as the scene progresses. To address this issue and enhance the extraction of spatial-audio features, we introduce **spatial-audio cross-attention module**, a cross-attention mechanism that merges spatial feature embedding $f(\mu_c)$ of the canonical 3D Gaussians with subsequent audio features, capturing how the input speech audio affects the movement of the 3D Gaussians. The spatial-audio cross-attention module comprises $L$ sets of cross-attention layer $\mathcal{T}_{CA}(\cdot)$ and feed-forward layer

$FFN(\cdot)$, each interconnected with skip connections. The module is formulated as:

$$z_n^0 = f(\mu_c),  \quad (10)$$

$$z_n'^l = \mathcal{T}_{CA}(z_n^{l-1}, a_n) + z_n^{l-1}, \quad l = 1...L, \quad (11)$$

$$z_n^l = FFN(z_n'^l) + z_n'^l, \quad l = 1...L, \quad (12)$$

whereby the cross-attention between the spatial feature $f$ and the audio feature $a_n$ of the $n$-th image frame is computed. As a result, the output feature $z_n^L$ successfully amalgamates audio features with the rich facial details captured by each 3D Gaussian. This cross-attention module offers a more nuanced and stable method of feature combination than simple concatenation or multiplication, as the module reforms the spatial-aware facial features with respect to the subsequent audio features, taking into account the dynamic variability inherent in each 3D Gaussian.

*4.3.2 Disentanglement of speech-related motion.* When synthesizing a talking head, the corresponding speech audio does not account for all the intricate and diverse facial movements. Subtle expressions like eye blinks and facial wrinkles, along with external factors such as hair movement and variations in lighting, do not directly correlate with input speech audio. Thereby, it is crucial to separate the non-verbal motions and scene variations when mapping speech audio to the 3D Gaussian deformation. In this section, we address this challenge by introducing additional input conditions that capture non-verbal motions, allowing us to disentangle speech-related motion from the monocular video.

Following previous works [23, 38], we first apply explicit eye blinking control with the eye feature $e$. Specifically, we employ AU45 from the Facial Action Coding System [9] to describe the degree of the eye blink, and utilize a sinusoidal positional encoding in order to match the input dimensions. Additionally, we integrate the camera viewpoint as an auxiliary input to disentangle non-verbal scene variations. While we formulate the framewise camera $\pi_n$ as facial viewpoints, the typical video is recorded with a static camera while the head undergoes continuous movement. Consequently, variations in the portrait image, such as hair displacement and lighting changes, occur independently of the speech audio. Hence, we employ a facial viewpoint embedding $v$ as an additional input condition to disentangle these non-auditory scene fluctuations. $v_n$ is an embedding vector obtained by mapping the extrinsic camera pose $\pi_n$ to a small MLP to have the same dimensionality as the other inputs. Finally, we discovered that using a single null-vector ($\emptyset$) for all frames promotes consistency as a global feature across video frames. We incorporate this null-vector as an additional input for our cross-attention network. Thus, we reformulate (11) as:

$$z_n'^l = \mathcal{T}_{CA}(z_n^{l-1}, \{a_n, e_n, v_n, \emptyset\}) + z_n^{l-1}, \quad l = 1...L. \quad (13)$$

In Fig. 4, we visualize the attention scores for each input in order to demonstrate the efficacy of disentangling audio-related motion. Further details on the network structure and visualization procedure are provided in the supplementary file.

*4.3.3 Audio-conditioned deformation of 3D Gaussian.* The final deformation network takes the spatially-aware audio features encoded in each 3D Gaussians in order to compute the deformation of position, rotation, and scaling. We define the set of MLP regressors

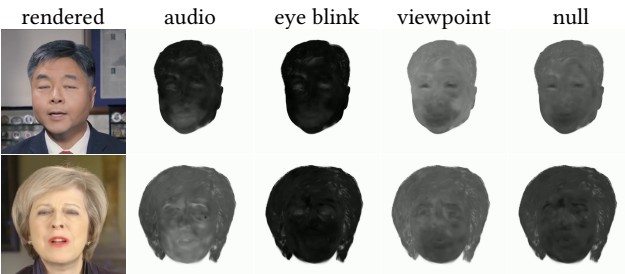

rendered    audio    eye blink    viewpoint    null

**Figure 4: Illustration of attention score distributions across different modalities for two individuals. From left to right: the original rendered image, attention scores responsible for audio cues, eye blink dynamics, head orientation (facial viewpoint), and temporal consistency (null), respectively.**

$\mathcal{F}_{\text{deform}}(\cdot)$ in order to predict the offsets of each Gaussian attributes, such as:

$$\{\Delta\mu_n, \Delta s_n, \Delta r_n, \Delta SH_n, \Delta\alpha_n\} = \mathcal{F}_{\text{deform}}(z_n^L). \quad (14)$$

## 4.4 Training

*4.4.1 Stage-wise optimization.* 3DGS [21] showed that the quality of reconstruction is influenced by the initialization of 3D Gaussians. Similarly, the training of the deformation field should also be conducted using a proper initialization of the canonical facial shape. To this end, we employ a two-stage training approach.

In the first stage, **canonical stage**, we first reconstruct the mean shape of the talking face, by optimizing the positions of 3D Gaussians and the multi-resolution triplane. Instead of the conventional initialization using structure from motion (SFM) points, we opt to utilize the 3D coordinates of the mesh vertices from fitting 3D morphable models. Note that the 3DMM fitting of each frame involves no extra preprocessing, as this is a necessary part of obtaining the camera parameters of the talking face and is widely adopted in NeRF-based talking face synthesis works [16, 23, 38]. The static image of the canonical talking head is rasterized via:

$$\hat{I}_{\text{can}} = R(\mathcal{G}_{\text{can}}; \pi_n). \quad (15)$$

This is followed by the **deformation stage**, where we optimize the whole network, from which we learn the cross-attention deformation network. For each frame, the dynamic talking head video frame can be rendered as:

$$\hat{I}_n = R(\mathcal{G}_{\text{deform},n}; \pi_n). \quad (16)$$

*4.4.2 Loss Functions.* For the **canonical stage** for a static shape of talking head, we follow the original 3DGS implementation [21] and utilize a combination of L1 color loss $\mathcal{L}_1$ and a D-SSIM term $\mathcal{L}_{\text{D-SSIM}}$. Following previous audio-driven NeRF works [16, 23, 38], we also utilize LPIPS [52] loss $\mathcal{L}_{\text{lpips}}$ to capture sharp details. For a given input frame $I$, the overall loss function of the **canonical stage** is denoted as $\mathcal{L}_{\text{can}} = \mathcal{L}_{L1} + \lambda_{\text{lpips}}\mathcal{L}_{\text{lpips}} + \lambda_{\text{D-SSIM}}\mathcal{L}_{\text{D-SSIM}}$. During the **deformation stage**, we employ an additional loss function on the lip area of the talking head. Specifically, we apply a reconstruction loss for the image patch obtained by cropping where the lips are located based on the facial landmarks [2]. Thus, the total loss function for the **deformation stage** can be formulated as

**Table 1: Quantitative comparison under the *self-driven* setting.**

| Methods | PSNR ↑ | SSIM ↑ | LPIPS ↓ | FID ↓ | LMD ↓ | AUE ↓ | Sync ↑ | CSIM ↑ | Training Time ↓ | FPS↑ |
|---|---|---|---|---|---|---|---|---|---|---|
| Ground Truth | N/A | 1 | 0 | 0 | 0 | 0 | 8.653 | 1 | N/A | N/A |
| Wav2Lip [32] | 30.461 | 0.911 | 0.024 | 33.074 | 4.458 | 1.761 | 9.606 | 0.887 | - | 19 |
| PC-AVS [56] | 21.958 | 0.699 | 0.053 | 42.646 | 4.619 | 1.875 | 9.185 | 0.519 | - | 32 |
| AD-NeRF [16] | 30.341 | 0.906 | 0.026 | 20.243 | 5.692 | 2.331 | 4.939 | 0.908 | 13h | 0.13 |
| RAD-NeRF [38] | 30.703 | 0.915 | 0.026 | 26.238 | 3.142 | 2.196 | 5.757 | 0.911 | 3h | 32 |
| ER-NeRF [23] | 31.673 | 0.919 | 0.014 | 19.829 | 3.003 | 1.974 | 5.976 | 0.922 | 1h | 34 |
| **GaussianTalker*** | 32.269 | 0.930 | 0.016 | 8.626 | 2.932 | 1.920 | 6.443 | 0.933 | 1h | 121 |
| **GaussianTalker** | 32.423 | 0.931 | 0.018 | 8.626 | 2.932 | 1.920 | 6.554 | 0.932 | 1.5h | 98 |

**Table 2: Quantitative comparison under the *cross-driven* setting. We extract two audio clips from SynObama demo [37] to drive each method and compare lip synchronization.**

| | Testset A | | | Testset B | | |
|---|---|---|---|---|---|---|
| Methods | Sync↑ | LMD↓ | AUE↓ | Sync↑ | LMD↓ | AUE↓ |
| Ground Truth | 7.850 | 0 | 0 | 6.976 | 0 | 0 |
| Wav2Lip [32] | 8.272 | 7.102 | 2.023 | 7.907 | 5.591 | 3.164 |
| PC-AVS [56] | 8.408 | 7.731 | 2.212 | 7.592 | 6.230 | 3.123 |
| AD-NeRF [16] | 5.128 | 18.986 | 3.654 | 5.109 | 9.221 | 3.266 |
| RAD-NeRF [38] | 5.126 | 12.485 | 3.611 | 4.497 | 7.760 | 3.447 |
| ER-NeRF [23] | 4.694 | 12.477 | 3.779 | 4.822 | 7.698 | 3.287 |
| **GaussianTalker** | 5.356 | 12.702 | 3.663 | 5.413 | 7.812 | 3.265 |

$\mathcal{L}_{\text{deform}} = \mathcal{L}_{\text{can}} + \lambda_{\text{lip}}\mathcal{L}_{\text{lip}}$. Note that the deformed 3D Gaussians are directly splatted onto the combined background and torso image, in order to render the head with the background and torso, a common technique that prevents noise around the facial contours [23, 38]. A more detailed explanation of this technique can be found in the supplementary file.

## 5 EXPERIMENTS

### 5.1 Experimental Settings

*5.1.1 Dataset and pre-processing.* For each target subject, we require several minutes of talking portrait video with a corresponding audio track for training. Specifically, the datasets are obtained from publicly-released video datasets utilized in previous NeRF-based works [16, 26, 34, 48], averaging 6,000 frames for each video at 25 fps. We also perform experiments on selected video clips sourced from the HDTF dataset. [54]. Each portrait video is cropped and resized to $512 \times 512$, apart from the Obama video, which is of the resolution $450 \times 450$. We split each video into train and test sets at a ratio of 10:1, following the pre-processing steps introduced in AD-NeRF [16].

*5.1.2 Comparison baselines.* We comparatively evaluate our proposed GaussianTalker framework against recent NeRF-based approaches tackling the same task. We introduce two variants of our method: the full model **GaussianTalker** with $L = 2$ cross-attention layers and a lightweight version, **GaussianTalker***, with $L = 1$ layer. Our method is compared with the recent NeRF-based approaches that address the same problem settings. We utilize three models as baselines: AD-NeRF [16], RAD-NeRF [38], and ER-NeRF [23]. For fair comparison, we implement each method by utilizing the torso part from the ground-truth frames. Additionally, we include a comparison with one-shot 2D talking head models, such as Wav2Lip [32] and PC-AVS [56], to provide a wide range of comparisons.

### 5.2 Quantitative Evaluation

*5.2.1 Comparison settings and metrics.* Following previous works [23, 38], our comparisons are structured into two distinct settings: **self-driven** and **cross-driven**. In the **self-driven** setting, we evaluate the accuracy of head reconstruction for a particular identity using the test subset. We employ several reconstruction metrics including peak signal-to-noise ratio (**PSNR**), structural similarity index measure (**SSIM**), and learned perceptual image patch similarity (**LPIPS**). Notably, these metrics are exclusively measured on the facial region. We also measure realism of the reconstructed face using Fréchet Inception Distance (**FID**) [17] and identity preservation of the animated video using Cosine Similarity of Identity Embedding (**CSIM**) [19].

For the **cross-driven** setting, all methods are driven by entirely unrelated audio tracks to evaluate lip synchronization. The audio clips used in this setup were extracted from demos of Syn-Obama [37]. Due to the absence of ground-truth images, we assess lip sync accuracy with landmark distance (**LMD**) and SyncNet confidence score (**Sync**). We also employ action units error (**AUE**) to measure the precision of facial movements. Finally, we compare the **training time** and frames-per-second (**FPS**) as measures to evaluate the efficiency of each method.

*5.2.2 Self-driven evaluation.* The **self-driven** evaluation results are presented in Tab. 1. Note that Wav2Lip [32] scores for PSNR, SSIM and LPIPS are not valid as it takes ground truth images as input. While the one-shot 2D-based methods, Wav2Lip and PC-AVS generate results with high synchronization scores, they fall short in the faithful reconstruction, showing low PSNR and LPIPS scores. Benefiting from the 3DGS representation, GaussianTalker achieves comparable image fidelity with significantly faster rendering speeds (over 120 fps for GaussianTalker*). Our method also shows the best scores in most metrics while reaching higher score than other NeRF-based baselines in Sync scores. The results show that our method can synthesize high lip-sync accurate 3D heads in real time rendering speeds.

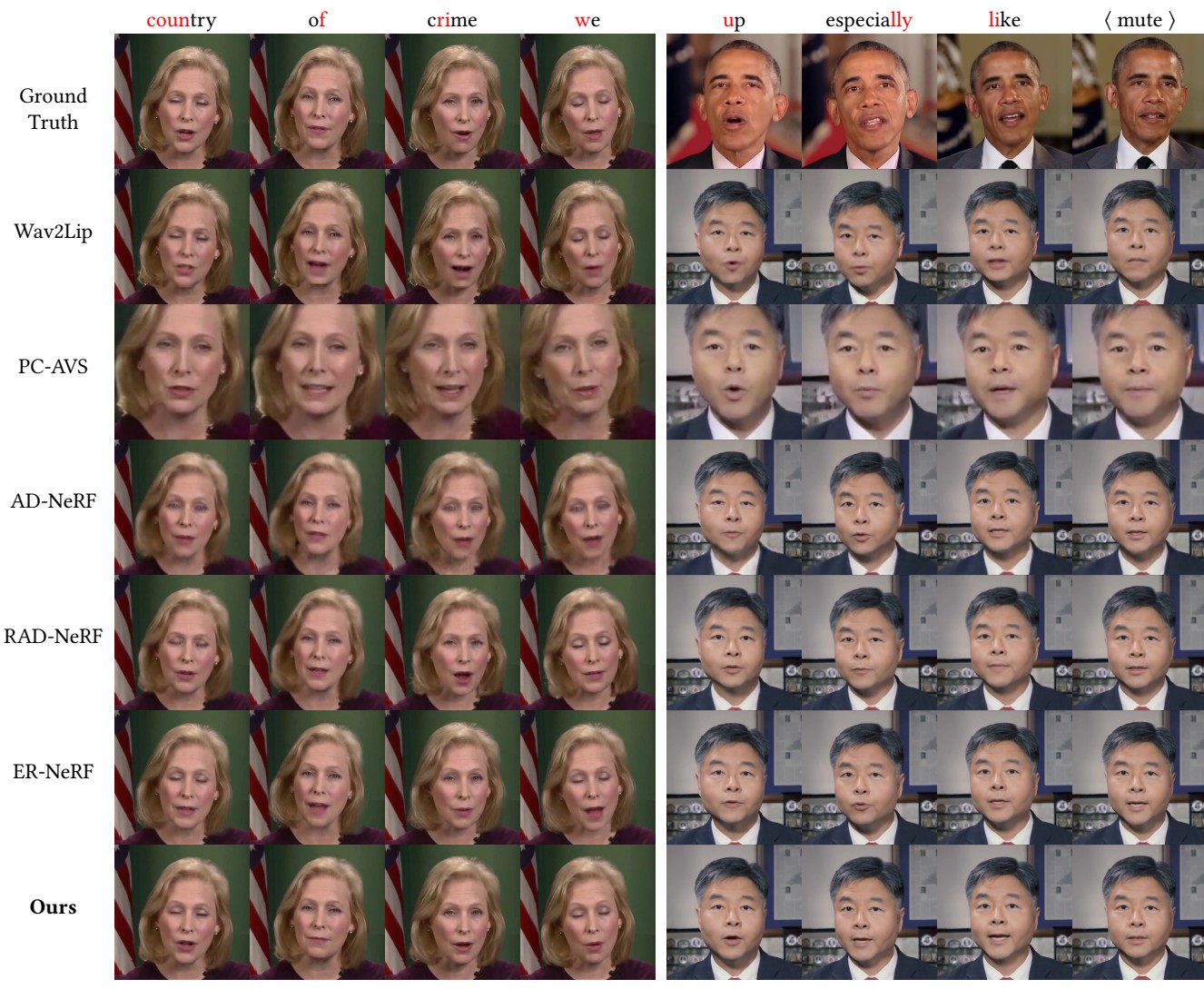

**Figure 5: Comparative visualization of lip synchronization across different audio-visual models. The sequence depicts the lip shape conforming to specific phonemes in the spoken words 'country', 'of', 'crime', 'we', 'up', 'especially', 'like', with the last frame showing a closed mouth ('mute').**

*5.2.3  Cross-driven evaluation.*  Results in Table 2 showcase successful lip movement synthesis with general audio input. GaussianTalker consistently exhibits the highest Sync score among NeRF-based methods, demonstrating its effectiveness in handling unseen audio for lip synchronization. These results highlight GaussianTalker's ability to generate high-fidelity 3D heads with real-time rendering speeds and accurate lip synchronization even with diverse audio inputs.

## 5.3  Qualitative Evaluation

In Fig. 5, we showcase results from self-driven and cross-driven experiments. We choose four key frames from each of the two experiment settings to compare the reconstruction quality and lip-sync accuracy. While 2D-based methods (Wav2Lip, PC-AVS) excel in lip synchronization, they for short of generating a faithful and consistent face when the head is rotated. AD-NeRF suffers

from blurry reconstructions due to its lack of eye blink control. RAD-NeRF and ER-NeRF, while demonstrating improved facial consistency, can exhibit discrepancies in lip synchronization and fail to capture hair movement during head rotations.

In contrast, GaussianTalker generates photorealistic images with intricate details in non-rigid regions like eyes and wrinkles. Our spatial-audio attention module effectively disentangles audio-driven motions from scene variations, enabling precise control of mouth movements. This capability allows our model to capture hair movement realistically when the head rotates, leading to superior overall head reconstruction fidelity. In order to comprehensively visualize the efficacy of our proposed method, we provide the rendered videos in the supplementary file. The provided supplementary video demonstrates impressive lip synchronization capabilities and high fidelity head reconstruction with realistic motion.

**Table 3: Ablation study results comparing various attribute configurations for embedding canonical 3D Gaussian attributes.**

| Method | PSNR ↑ | LPIPS ↓ | FID ↓ | LMD ↓ | Sync ↑ |
|---|---|---|---|---|---|
| Ground Truth | N/A | 0 | 0 | 0 | 8.935 |
| $s, r, SH, \alpha$ | 33.195 | 0.016 | 9.976 | 2.873 | 6.927 |
| $SH, \alpha$ | 33.299 | 0.014 | 9.808 | 2.891 | 6.853 |
| $r, s$ | 33.056 | 0.016 | 11.775 | 2.873 | 6.892 |
| random init. | 33.040 | 0.017 | 11.915 | 2.996 | 6.543 |

**Table 4: Ablation study on selection of deformed attributes.**

| Method | PSNR ↑ | LPIPS ↓ | FID ↓ | LMD ↓ | Sync ↑ |
|---|---|---|---|---|---|
| Ground Truth | N/A | 0 | 0 | 0 | 8.935 |
| $\Delta SH, \Delta \alpha$ | 32.746 | 0.021 | 44.933 | 3.179 | 6.694 |
| $\Delta\mu, \Delta r, \Delta s$ | 33.036 | 0.013 | 17.52 | 2.970 | 6.688 |
| $\Delta\mu, \Delta r, \Delta s, \Delta SH \Delta \alpha$ | 33.299 | 0.013 | 9.808 | 2.890 | 6.928 |

**Table 5: Ablation study on augmented input conditions.**

| Method | PSNR ↑ | LPIPS ↓ | FID ↓ | LMD ↓ | Sync ↑ |
|---|---|---|---|---|---|
| Ground Truth | N/A | 0 | 0 | 0 | 8.935 |
| w/o null-vec | 32.997 | 0.014 | 9.908 | 2.933 | 6.698 |
| w/o eye feature | 32.826 | 0.015 | 10.060 | 2.902 | 6.911 |
| w/o viewpoint | 31.866 | 0.019 | 13.231 | 3.052 | 6.563 |
| All (Ours) | 33.299 | 0.014 | 9.809 | 2.891 | 6.928 |

**Table 6: Ablation study on the effectiveness of stage-wise training.**

| Method | iter. | PSNR ↑ | LPIPS ↓ | FID ↓ | LMD ↓ | Sync ↑ |
|---|---|---|---|---|---|---|
| Ground Truth | - | N/A | 0 | 0 | 0 | 8.935 |
| w/o stage-wise | 500 | 26.063 | 0.072 | 66.629 | 3.446 | 1.348 |
| | 1000 | 26.478 | 0.064 | 56.890 | 3.344 | 4.007 |
| | 5000 | 32.676 | 0.016 | 14.026 | 2.971 | 6.602 |
| w/ stage-wise | 500 | 31.076 | 0.029 | 31.301 | 3.792 | 1.548 |
| | 1000 | 31.923 | 0.024 | 20.366 | 3.245 | 4.449 |
| | 5000 | 32.733 | 0.014 | 11.173 | 2.923 | 6.736 |

## 5.4 Ablation Study

In this section, we provide ablation studies to validate the efficacy of the design choices of our model. We also show detailed visualizations of the generated results in the supplementary material for better comparison.

*5.4.1 Attribute conditions for triplane.* Our proposed triplane encodes the facial information of the canonical 3D head learned by 3D Gaussians. The mechanism also enforces spatial relationships between Gaussians for better deformation. In Tab. 3, we demonstrate the effectiveness of this approach by conducting quantitative ablation on the selection of attributes that are conditioned on the embedding $f(\mu_c)$. We also provide results where all attributes are optimized separately following the original implementation, and the triplane is trained in the deformation stage. Utilizing only subsets of the Gaussian attributes show lower performance in lip synchronization and precision. Removing the attribute conditions during training leads to loss of spatial information embedded in the triplane embeddings, leading to a lack of facial cohesion during inference time.

*5.4.2 Selection of deformed attributes.* A major challenge of manipulating the Gaussians is the magnitude of the parameters that need to be controlled. While estimating offsets for only a subset of attributes could reduce computational load, it may compromise overall fidelity due to the lack of control. To address this, in Tab. 4, we investigate different selections of Gaussian attributes for deformation. Controlling only $SH$ and $\alpha$ makes the formulation similar to conditional NeRF-based works [16, 23, 38]. Because 3DGS is an explicit representation that specifies the 3D positions and shapes, only controlling the appearance attributes leads to loss of overall fidelity. However, only controlling attributes that make up the position and shape of 3D Gaussians show lower reconstruction accuracy. Deformation of all Gaussian attribute is crucial for the highest fidelity and superior lip synchronization.

*5.4.3 Disentanglement of audio-unrelated motion.* We also investigate the significance of using augmented conditions, such as eye blink, facial viewpoint, and null-vector. We evaluate the influence of additional conditions on image fidelity and lip synchronization by selectively removing them during training (Table 5). The lower reconstruction scores are attributed to the low lip-sync accuracy due to entanglement of verbal motion and scene variations unrelated to audio. In the supplementary material, we also visualize the attention scores of each comparison experiment for detailed analysis.

*5.4.4 Stagewise optimization.* In Fig. 6, we investigate the importance of employing a separate **canonical stage**. We opt to optimize the whole architecture by training each of the module simultaneously from scratch. While the final generated results show similar performance, optimizing the coarse facial geometry before training the deformation network results in faster optimization of the whole methodology.

## 6 CONCLUSION

In this work, we have proposed GaussianTalker, a novel framework for real-time pose-controllable 3D talking head synthesis, leveraging the 3D Gaussians for the head representation. Our method enables precise control over Gaussian primitives by conditioning features extracted from a multi-resolution triplane. Additionally, the integration of a spatial-audio cross-attention module facilitates the dynamic deformation of facial regions, allowing for nuanced adjustments based on audio cues and enhancing verbal motion disentanglement. Our method is distinguished from prior NeRF-based methods by its superior inference speed and high-fidelity results for out-of-domain audio tracks. The efficacy of our approach is validated by quantitative and qualitative analyses. We look forward to enriched user experiences, particularly in video game development, where real-time rendering capabilities of GaussianTalker promise to enhance interactive digital environments.

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
