# OpenReview forum: "GaussianTalker: Real-Time Talking Head Synthesis with 3D Gaussian Splatting"
_acmmm.org/ACMMM/2024/Conference — MM2024 Poster_

### Official Review · Reviewer_JYtU · 2024-05-21

**Rating:** 4
**Confidence:** 2

**Summary:**

This paper designs a real-time generation of high-quality, pose-controllable talking heads. The author proposes the GaussianTalker framework to address the challenges of audio driven 3DGS rendering technology in real-time applications. This framework encodes 3D Gaussian attributes into shared implicit feature representations and fuses them with audio features to control head deformation during audio synchronization. This design utilizes the spatial information of the head and enhances the interaction between adjacent points. This framework uses a spatial-audio attention module to stably predict the frame-wise offset of each Gaussian attribute, which improves the ability to handle a large number of Gaussian distributions and their complex parameters compared to previous direct concatenation or multiplication methods. The experimental results show that GaussianTalker surpasses previous methods in facial fidelity, lip synchronization accuracy, and rendering speed, and the effectiveness of each design component has been verified through ablation studies.

**Strengths:**

The paper proposes a novel framework for real-time generation of pose-controllable talking heads, achieved through audio driven 3DGS technology. The key innovation of the research lies in effectively solving the problem of directly controlling 3DGS with audio by encoding 3D Gaussian attributes into shared implicit feature representations and fusing them with audio features. At the same time, spatial-audio attention module is introduced to predict the frame-wise offset of each Gaussian attribute, thereby improving the stability and region specific deformation ability of the system. Compared to NeRF based methods, GaussianTalker not only performs better in facial realism, lip synchronization accuracy, and rendering speed, but also successfully overcomes the challenges of a large number of Gaussians and fine parameter tuning.

The advantage of this paper is that it provides a theoretically reasonable and thoroughly empirically evaluated new method for audio driven 3D heads. Through detailed quantitative comparison and ablation research, the paper demonstrates its significant advantages over current advanced methods such as AD-NeRF, RAD-NeRF, and ER-NeRF, including higher FPS and better fidelity. In addition, the paper is suited to the ACM MM community, as it combines technologies from multiple modalities such as 3D reconstruction and audio processing, promoting the development of real-time interactive media and virtual character expression.

**Limitations:**

The Gaussian Talker framework proposed in the paper demonstrates a novel and promising approach aimed at achieving real-time, high fidelity, and controllable speaker head synthesis. It utilizes audio driven 3DGS technology to promote progress in this field. However, the paper did not clearly explain the principles of some of the designs in the framework, and some of the figures and tables need to be modified for readers to understand.

1）	The conclusion mentioned in line 133 of the paper, "The cross attention offers a more stable approach of manipulating the substantial number of Gaussians and their intricate parameter space, compared to concatenation or multiplication as in previous works" lacks experimental or theoretical evidence to support it.

2）	Line 507 of the paper mentions "we discovered that using a single null-vector $(∅)$ for all frames promotes consistency as a global feature across video frames". The paper did not explain in detail why null-vector $(∅)$ can improve consistency. And in Figure 4, why there are differences in the attention scores of the null-vector $(∅)$ in different regions of the head, and why the null-vector $(∅)$  is responsible for temporal consistency.

3）	Some details of the graph need to be further optimized to improve its clarity and accuracy:

a)	In Figure 2, to ensure that readers have an accurate understanding of Gaussian Talker, it is recommended to indicate the parameters μ of $\mathcal{G}_{can}$ in the figure to avoid any possible misunderstandings.

b)	For Figure 5, in order to more intuitively demonstrate the shortcomings of other methods, it is recommended to add appropriate markings (refer to ER-NeRF or Figure A6 in the supplementary materials of the paper). For example, the statement "RAD-NeRF and ER-NeRF,... fail to capture hair movement during head rotations." mentioned in line 797 of the paper does not seem easy to observe from Figure 5.

c)	In Figure 5, the GT image labeled "mute (a closed mouth)" actually shows an open mouth.

4）	Regarding the details of the experimental section:

a)	The paper mentions in line 684 that the PSNR and LPIPS of Wav2Lip are invalid, but relevant comparisons are made later in the text. To avoid confusion for readers, it is recommended to reorganize the content.

b)	To improve the readability of the table, it is recommended to refer to Table A1 in the supplementary materials and clearly indicate the maximum values of each indicator in the table. This will help readers quickly identify the best performance among various indicators.

c)	In the last row of Table 4, a comma is missing between $Δ𝑆𝐻$ and $Δα$.

5）	In order to comprehensively evaluate the performance of the model, it is recommended to conduct more extensive cross driven experiments, such as using audio from different languages to drive the model (refer to GeneFace++), to explore the adaptability and generalization ability of the model.

6）	In line 478 of the paper, "... as the module reforms the spatial-aware facial features with respect to the subsequent audio features, ...", What is "spatial aware medical features" and is it "feature embedding $f(μ_c)$"?

**Suitability:**

3

---

### Official Review · Reviewer_TcTZ · 2024-05-22

**Rating:** 3
**Confidence:** 3

**Summary:**

This paper presents GaussianTalker, a framework for real-time generation of pose-controllable talking heads. It employs the fast rendering capabilities of 3D Gaussian Splatting (3DGS) while addressing the challenges of directly controlling 3DGS with speech audio.

**Strengths:**

1. This paper presents an audio-conditioned 3D Gaussian Splatting framework for real-time 3D-aware talking head synthesis.
2. This paper reformulates the 3D Gaussian representation with a feature volume representation to enforce spatial consistency between adjacent Gaussians.
3. This paper integrates cross-attention mechanisms between audio and spatial features to improve stability and ensure region-specific deformation over a significant number of Gaussians.

**Limitations:**

About the Experimental Section:
1. The best results in the table should be highlighted in bold.
2. In Section 5.2, how many AUs were used for AUE? How many landmarks were used for LMD? Relevant literature must be referenced for these evaluation metrics.
3. Details should be highlighted or enlarged in the qualitative comparison results.
4. In Tables 3 and 4 of the ablation experiments, the differences in results based on parameter selection are not significant. Can further analysis and explanation be provided? From the tables, it can be observed that using only “SH and a” yields the best results. Does this mean that only “SH and a” should be selected?

About the Method:
1. Can the proposed 3DGS-based method generate heads from other viewpoints? Please show if it can.
2. Please describe in detail the face decomposition process in Section 4.3.2. It seems that a series of methods, including loss functions, are required to achieve the results shown in Figure 4, but this section is described rather generally.
3. Besides the audio decomposition part, the differences with GaussianHead and MonoGaussianAvatar [1] should be clarified. Except for the audio decomposition part, the model of this paper is similar to GaussianHead and MonoGaussianAvatar, especially MonoGaussianAvatar, including the extracted face parameters.
[1] Yufan Chen, Lizhen Wang, Qijing Li, Hongjiang Xiao, Shengping Zhang, Hongxun Yao, and Yebin Liu. 2023. Monogaussianavatar: Monocular gaussian point-based head avatar. arXiv preprint arXiv:2312.04558 (2023).

**Suitability:**

2

---

### Official Review · Reviewer_JGxV · 2024-05-22

**Rating:** 4
**Confidence:** 3

**Summary:**

This paper introduces an innovative framework designed for real-time talking head synthesis. It utilizes 3D Gaussian Splatting (3DGS) for fast rendering while addressing the challenge of synchronizing deformation with speech audio. A feature volume representation and cross-attention mechanism are proposed for enhancing the consistency of spatial information and the relevance of audio and spatial features, respectively. The proposed methods achieve good performances on several datasets.

**Strengths:**

1.	The paper offers a new path for introducing 3DGS into talking head synthesis, realizing impressive rendering speeds and quality.
2.	The paper is well-structured, logical, and easy to follow.

**Limitations:**

1.	The proposed method relies heavily on the dataset used for training thus making it difficult to generalize to different speaker profiles and styles.
2.	Qualitative and quantitative comparisons should be boxed for the area to be compared and bold the best results separately.
3.	Sharp teeth and hollow areas inside the mouth require finer reconstruction, while more efficient point cloud pruning methods should be used to improve optimization efficiency.
4.	More quantitative experiments are needed to compare the cross-attention mechanism with concatenation or multiplication.

**Suitability:**

3

---

### Official Review · Reviewer_1ncv · 2024-06-01

**Rating:** 5
**Confidence:** 3

**Summary:**

This work represents the first time to leverage the 3D Gaussian representation to exploit its fast scene modeling capability for audio-driven dynamic facial animation. It first employs triplane to extract feature embeddings for each 3D Gaussian position. Then, it uses the cross attention for fusing feature embeddings and audios. Experments shows that GaussianTalker achieves on par with or better results at much higher FPS compared to SOTAs.

**Strengths:**

1. Clear idea for integrating the GS into Talking Head, which brings the high inference time and PSNR.
2. High quality for paper and its supplementary material.
3. Sufficient experiments and clear illustration for proving the proposed method's effectiveness.

**Limitations:**

I don't have any problem with this paper due to its high quality and breakthrough in talking head synthesis.

**Suitability:**

3

---

### Meta-Review · Area_Chair_Lcok · 2024-06-30

**Recommendation:** Accept (Poster)
**Confidence:** 4

**Metareview:**

This paper introduces a framework designed for real-time talking head synthesis by means of 3D Gaussian Splatting (3DGS) for fast rendering.

The paper has mostly positive reviews (WA, BA, BR, BA). Reviewers recognize several positive points:
- A convincing and novel framework where gaussian splatting are conditioned with audio features. The framework looks innovative and effective.
- Paper is of high quality and clear, technically sound and experiments thorough

The main drawback identified by reviewers are mostly related to unclear technical details, minor generation defects and clarity of some parts. The rebuttal includes discussions and additional results that seem clear and in support of the design choices.

The AC believes that positive aspects overcome negative ones in this case. The framework looks innovative and minor defects are reasonable. As suggested by Reviewers, the paper can be a good contribution to the venue.